Manuscript prepared for Geosci. Model Dev.
with version 2014/09/16 7.15 Copernicus papers of the LaTeX class copernicus.cls.
Date: 16 December 2016

# Half a degree Additional warming, Prognosis and Projected Impacts (HAPPI): Background and Experimental Design

Daniel Mitchell[1], Krishna AchutaRao[2], Myles Allen[1,3], Ingo Bethke[4],
Andy Ciavarella[5], Piers Forster[6], Jan Fuglestvedt[7], Nathan Gillett[8],
Karsten Haustein[1], William Ingram[3,5], Trond Iversen[9], Viatcheslav Kharin[8],
Nicholas Klingaman[10], Neil Massey[1], Erich Fischer[11],
Carl-Friedrich Schleussner[12,13], John Scinocca[8], Øyvind Seland[9],
Hideo Shiogama[14], Emily Shuckburgh[15], Sarah Sparrow[16], Dáithí Stone[17],
Peter Uhe[16,1], Beyerle Urs[11], David Wallom[16], Michael Wehner[17], and
Rashyd Zaaboul[18]

[1]Environmental Change Institute, School of Geography and the Environment, Oxford University, Oxford, UK.
[2]Centre for Atmospheric Sciences, Indian Institute of Technology Delhi, New Delhi 110016. India.
[3]Atmospheric, Oceanic and Planetary Physics (AOPP), Oxford University, Oxford, UK.
[4]Uni Research Climate, Bjerknes Centre for Climate Research, Bergen, Norway
[5]Met Office Hadley Centre for Climate Science and Services, Exeter, UK.
[6]School of Earth and Environment, University of Leeds, Leeds, UK.
[7]Center for International Climate and Environmental Research - Oslo (CICERO), PO Box 1129 Blindern, 0318 Oslo, Norway.
[8]Canadian Centre for Climate Modelling and Analysis, Environment and Climate Change Canada, University of Victoria, Victoria, V8W 2Y2, Canada.
[9]Norwegian Meteorological Institute, Oslo, Norway
[10]National Centre for Atmospheric Science - Climate, Department of Meteorology, University of Reading, Reading, UK.
[11]ETH Zurich, Institute for Atmospheric and Climate Science, Zurich, Switzerland.
[12]Climate Analytics, Potsdam, Germany.
[13]Potsdam Institute for Climate Impact Research, 14473 Potsdam, Germany.
[14]Center for Global Environmental Research, National Institute for Environmental Studies,16-2 Onogawa, Tsukuba, Ibaraki 305-8506, Japan.
[15]British Antarctic Survey (BAS), High Cross, Madingley Road, Cambridge, UK.
[16]Oxford e-Research Centre (OeRC), University of Oxford, Oxford, UK.
[17]Lawrence Berkeley National Laboratory, Berkeley, CA, USA.
[18]International Center for Biosaline Agriculture, PO Box 14660 Dubai, UAE.

*Correspondence to:* Daniel Mitchell (d.m.mitchell@bristol.ac.uk)

**Abstract.**

The Intergovernmental Panel on Climate Change (IPCC) has accepted the invitation from the UN-FCCC to provide a special report on the impacts of global warming of 1.5°C above pre-industrial levels and on related global greenhouse gas emission pathways. Many current experiments in, for example, the Coupled Model Inter-comparison Project (CMIP), are not specifically designed for informing this report. Here, we document the design of the Half a degree Additional warming, Projections, Prognosis and Impacts (HAPPI) experiment. HAPPI provides a framework for the generation

of climate data describing how the climate, and in particular extreme weather, might differ from the present day in worlds that are 1.5°C and 2.0°C warmer than pre-industrial conditions. Output from participating climate models includes variables frequently used by a range of impact models. The key challenge is to separate the impact of an additional approximately half degree of warming from uncertainty in climate model responses and internal climate variability that dominate CMIP-style experiments under low emission scenarios.

Large ensembles of simulations (>50 members) of atmosphere-only models for three time slices are proposed, each a decade in length; the first being the most recent observed 10-year period (2006-2015), the second two being estimates of the a similar decade but under 1.5 and 2°C conditions a century in the future. We use the Representative Concentration Pathway 2.6 (RCP2.6) to provide the model boundary conditions for the 1.5°C scenario, and a weighted combination of RCP2.6 and RCP4.5 for the 2°C scenario.

## 1 Introduction

In its Paris Agreement, the parties to the United Nations Framework Convention on Climate Change (UNFCCC) has established a long-term temperature goal for climate protection of "holding the increase in the global average temperature to well below 2 °C above pre-industrial levels and pursuing efforts to limit the temperature increase to 1.5 °C above pre-industrial levels, recognizing that this would significantly reduce the risks and impacts of climate change" UNFCCC (2015). Such an agreement has naturally received interest from the academic community, with numerous authors commenting on this outcome (e.g. Hulme, 2016; Peters, 2016; Rogelj and Knutti, 2016; Mitchell et al., 2016b; Anderson and Nevins, 2016; Boucher et al., 2016; Schleussner et al., 2016). However, the body of research assessing impacts under a 1.5°C world is small compared to higher emission scenarios studies (James et al., accepted), though there are notable exceptions (Fischer and Knutti, 2015; Schleussner et al., 2015). It has been argued that current coordinated international climate modeling experiments, such as the Coupled Model Intercomparison Project (CMIP5) (Taylor et al., 2012), may not be best suited to address this question, and so we need dedicated climate experiments (Mitchell et al., 2016b).

HAPPI is proposed to provide a framework to assess the impacts of a 1.5°C world, and the impacts avoided from higher degree worlds, such as 2°C. As argued in Mitchell et al. (2016b), assessment of the impacts of a 1.5°C world requires large sets of simulations in order to adequately sample the extreme weather that often is associated with the highest climate-related impacts and risks, and it also requires simulations under steady forcing conditions in order to address the 1.5°C target. Figure 1 shows a schematic of how HAPPI differs from scenario-based approaches, such as CMIP. The more traditional scenario-based approach (top panel) starts with either an emission scenario, such as those used in CMIP3 (Special Report on Emissions Scenarios; SRES)(Nakicenovic and Swart,

2000), or a pathway to reach a certain radiative forcing by 2100, such as those used in CMIP5 (Representative Concentration Pathway; RCP)(Van Vuuren et al., 2011). As uncertainty increases with time, and is dominated by responses and variability in CMIP-style experiments, as illustrated in Figure 1 (upper panel), such experiments are not ideal to inform assessments of impacts at specific levels of warming such as 1.5°C or 2°C, let alone the difference between two such warming levels. For example, the lowest CMIP5 scenario, the RCP2.6, shows a median global mean temperature increase of 1°C above 1986-2005 levels, with a likely range between 0.3 and 1.7°C over the CMIP5 model ensemble (IPCC, 2013). This range includes 1.5°C and 2°C warming above pre-industrial levels, which introduces some issues into the assessment of differences in impacts of these warming levels based on such a model ensemble. Some studies have used methodologies with CMIP5 models that partially address this issue, for instance Fischer and Knutti (2015) pick 20 year periods from transient simulations centered on a specific global mean temperature threshold. Such a method has advantages over the HAPPI method in that it taps into the wealth of model integrations already performed in CMIP, but also that it samples SST variability across the board (the atmospheric models are coupled to interactive oceans)[1]. However, it also adds an extra level of complexity in that there is a large spread in timing for when transient CMIP models cross 1.5C, and difference forcings will be at play during different times. One example is ozone hole recovery and the implications for southern hemisphere circulation patterns, which are likely to be different if, for example a model crosses 1.5C in 2030 rather than 2050 (e.g. Son et al., 2010). It is also harder to calculate a robust return period from transient simulations, because contiguous data will only be be consistent with a global mean temperature threshold for a short period of time.

The parties to the UNFCCC have chosen to frame their goals for climate protection in terms of a global temperature response, rather than an emission scenario. As such, the UNFCCC is not asking for the risks associated with emission scenarios that is "likely" to maintain temperatures below 1.5°C (or some other criterion): it is asking about the risks associated with 1.5°C warming per se, irrespective of what emission path is followed to achieve it (emission paths being addressed in the second challenge). As such, the global response is where the HAPPI design starts, tracing through to regional extreme weather and potential impacts.

## 2  Experimental Design

The experiments under HAPPI are designed to be as similar as possible in experimental design as current (or proposed) climate experiments, notably the International CLIVAR Climate of the 20th Century Plus Detection and Attribution (C20C+ D&A) project (Gillett et al., 2016; Folland et al., 2014). Synergies between the experiments allows to minimize the additional computational time required from modeling centers. The core experiments will be driven with a spectrum of different

---

[1]This is explicitly addressed in Section 2 as a sensitivity test to the HAPPI design.

Table 1: Table of models that will likely contribute to HAPPI with specifications and expected number of simulated model years per experiment tier. Regional Climate Models (RCMs) are also listed. In addition to the simulations detailed here, modeling centres will run five ensemble members of 1959 to 2015 conditions for bias-correction purposes.

| Model | Hor. Resolution | Tier 1 | Tier 2 | RCM | References |
|---|---|---|---|---|---|
| CAM4 | 2×2° | 15,000 | 0 | N | Neale et al. (2013) |
| CAM5.1-0.25degree | 25×25 km | 90 | 0 | N | Wehner et al. (2014) |
| CAM5.1-1degree | 1.25×0.94° | 3000 | 6000 | N | Neale et al. (2010) |
| CanAM4 | T63 | 1500 | 0 | N | von Salzen et al. (2013) |
| HadAM3P | 1.88×1.25° | 30,000 | 30,000 | Y | Massey et al. (2014) |
| HadGEM3 | N216 | 1500 | 0 | N | Walters et al. (2016) |
| MetUM-GOML2 | 1.875×1.25° | 0 | 450 | N | Hirons et al. (2015) |
| | | | | | Walters et al. (2016) |
| MIROC5 | 150 × 150 km | 3000 | 0 | N | Shiogama et al. (2014) |
| MPI-ECHAM6.3 | T63 | 3000 | 0 | Y | - |
| NorESM1_Happi | 1.25×0.94° | 3750 | 2000 | N | Bentsen et al. (2013) |
| | | | | | Kirkevåg et al. (2013) |
| | | | | | Iversen et al. (2013) |

leading atmosphere-only Global Circulation Models (GCMs), the initial participants of which are listed in Table 1. By using atmosphere-only models instead of fully coupled models, we are able to generate larger ensemble sizes (due to decreased computational cost) while providing more accurate regional climate projections (He and Soden, 2016). Boundary conditions for the models are taken from the CMIP5 experimental design and from models that participated in that initiative.

There are two tiers of experiments, intended to characterize various climate scenarios, as well as uncertainties in the specifications of the temperature-based scenarios.

## 2.1 Tier 1 Experiments

Three core experiments are proposed:

1. Current decade conditions (2006-2015 50- to 100-member ensembles).

2. 1.5°C warmer than preindustrial (1861-1880) conditions (50- to 100-member ensembles) relevant for the 2106-2115 period.

3. 2.0°C warmer than preindustrial (1861-1880) conditions (50- to 100-member ensembles) relevant for the 2106-2115 period.

Each simulation within an experiment differs from the others in its initial weather state. The use of 50-100 10-year time slices provides 500-1000 years of data per experiment. Simulations are limited to 10 years in length because the observed ocean temperatures, upon which all HAPPI experiments are based, have been approximately constant during this period (at least within the context of the anthropogenic warming scales considered by HAPPI). However, 10-year periods should provide material for some analysis of multi-year events, such as droughts. The degree to which the output of the simulations can be used to estimate unbiased return values for a specific return period will depend on various aspects of the event, such as region and climate variable. In the extratropical summer, for instance, the 500-1000 years may be considered an unbiased sample, whereas in the tropics it may be important to acknowledge the absence of a major La Niña event during the 2006-2015 period.

**Current decade experiment:** Modeling centers will use observed forcing conditions as in the DECK AMIP design, including Sea Surface Temperatures (SSTs) and sea ice (Taylor et al., 2012). The 2006-2015 decade is chosen because it is our most recently observed period, but also because it contains a range of different SST patterns over the decade, allowing for an assessment of how the ocean conditions vary on inter-annual timescales. From 2017 onward, modeling centers will also have the option of simulating observed 2016 climate, thereby capturing the large El Niño event in 2015-2016. Note that the C20C project will also perform these experiments.

**The 1.5°C experiment:** It is difficult (without many climate-model-specific iterations) to explicitly design an emissions scenario that would lead to a world exactly 1.5°C warmer than preindustrial conditions. This is because the CMIP community are set up to use particular emission scenarios or RCP scenarios, rather than a scenario that leads to some chosen amount of warming. Here, we take 1.5°C to mean '1.5°C as measured as the near-surface air temperature', as is the formal definition of the transient climate response, rather than some mix of measuring systems (for instance surface ocean) that may have implications for the energy-budget (Richardson et al., 2016).

By chance, the average across climate model simulations submitted to CMIP5 under the RCP2.6 forcing scenario results in a global average temperature response at 1.55°C relative to preindustrial (2091-2100 relative to 1861-1880). Figure 2 shows the average and 5-95% spread in global mean temperature anomaly for all available CMIP5 models for the RCP2.6 scenario (dark blue). Within HAPPI, we assume that this amount of warming is sufficiently close to inform the call of the UN-FCCC on a special report on the "impacts of global warming of 1.5°C above pre-industrial levels" (UNFCCC, 2015), and thus HAPPI adopts the end-of-century anthropogenic radiative forcing conditions from the RCP2.6 emissions scenario. Specifically, forcing values for the year 2095 for GHG, aerosol and land use/cover changes are repeated for each of the years within the 1.5C decade. Natural radiative forcings, however, are set to the same values as in the current-decade experiment.

Projected SSTs are calculated by adding to the observed 2006-2015 SSTs a change in SST ($\Delta$SST) between the decadal-average of the modeled 2006-2015 period and the decadal-averaged of the modeled 1.5°C world over 2091-2100. Hence the SST patterns are still time-varying because they are

based on the 2006-2015 observations, but they have an additional warming added to them. As CMIP5 historical simulations stopped in 2005, the decadal average of the 2005-2015 SSTs is estimated from RCP8.5 simulations, as this is the scenario that is closest to observations over this period. The decadal average of the 2091-2100 SSTs is estimated from CMIP5 RCP2.6 simulations. The spread of these models is shown in Figure 2, of which 23 models have the required data (see Section 2.2 for more details on the individual patterns). The resulting multi-model average $\Delta$SST, used in the 1.5°C experiment, is shown in Figure 3. The global mean SST response is 1.02°C relative to the preindustrial period, with larger warming over land providing the global 1.55°C total. Because of the time period we use as our baseline (2006-2015) some of the so-called hiatus effect may bias our results cold, this will be partially compensated by the fact that our global mean temperature is 0.05°C higher than desired, but we also note that it is the difference between the 0.5°C warming in these relatively low emission scenarios that is important, rather than the exact magnitude.

Estimated sea ice is more problematic than estimated SSTs, because the CMIP-projected Arctic and Antarctic sea ice extents vary dramatically between models (Collins et al., 2013). In the Arctic, most climate models show a decrease at all longitudes in sea ice. In the Antarctic, the overall model responses show a similar decrease with equally variable projections. The CMIP5 climate models are also unable to capture the observed increases in Antarctic sea ice over the satellite era (Turner et al., 2013), leading to low confidence in their ability to predict future changes. As such, we use a different method to estimate sea ice under 1.5°C and higher scenarios, which is an adaptation of Massey (in prep). In short, we calculate an anomaly (from 1996-2015) for every month from 1996-2015 in both SSTs and sea ice from the OSTIA data set (Stark et al., 2007) and fit a linear relationship between SSTs and sea ice as a function of month and grid box. We use as the regressor the meridional average of SST grid boxes, within a hemisphere, at grid points where there is ice present at some point in time between 1996 and 2015 (i.e. the climatological monthly mean ice concentration for the grid box is non-zero). This represents temperature at that longitude under and near the ice edge, thereby minimizing poorly observed values in ice-covered regions. We use ice cover in an index gridbox as the regressand, and smooth the resultant field with a 500 km smoother. We then apply the sea ice-SST relationship to the 1.5°C experiment SST anomalies, to give a projected sea ice concentration anomaly. These anomalies are added on to the observed OSTIA data spanning the most recent decade. The absolute sea ice concentration fields, and anomalies from observations are given in Figure 4. This methodology has the added benefit that the SSTs and sea ice are consistent with each other in the HAPPI experiments.

**The 2°C experiment:** For the 2°C experiment, no analogous CMIP5 simulations are available. The RCP scenario resulting in the second coolest temperatures by the end of the 21st century is RCP4.5, which reaches ~2.5°C relative to preindustrial by the end of the 21st century (Fig. 2). Both RCP2.6 and RCP4.5 have 5-95% ranges that overlap a GMT of 2°C, and the mean of both scenarios are a similar distance from this threshold.

To calculate the future SST and sea ice conditions of a 2°C world we therefore take a weighted sum of the two RCP scenarios, W1 × RCP2.6 + W2 × RCP4.5. The weights are calculated such that the global mean temperature response is 2.05°C (i.e. exactly half a degree above the 1.55°C response from the 1.5°C experiment), and results in W1 = 0.41 and W2 = 0.59. These weights are used to calculate the SSTs and sea ice coverage using the same methodology as in the 1.5°C experiment.

The same weightings are applied to the radiative forcing of each well-mixed greenhouse gas (e.g. CO2, CH4, N2O, CFCs etc). Some concentrations do not scale linearly with radiative forcing, for instance $CO_2$ concentrations following a logithm, and the $CH_4$ and $N_2O$ concentrations follow a square root. All other concentrations are linearly related to the radiative forcing. A full list of these relationships is given by the IPCC (AR3, 2001). Natural forcings remain at the 1.5°C experiment

(and current-decade experiment) values. Land cover/use is represented in a discretised form in the climate models, and so cannot be interpolated. Meanwhile, the climate responses to anthropogenic aerosols and ozone concentrations (or, for some models, emissions of their precursors) do not follow a simple functional form, and in the case of aerosols this is further complicated by major differences in the spatial distributions of concentrations between the two RCPs. Considering that the parties to

the UNFCCC are most concerned about a $CO_2$-dominated warming, and is the dominant contributor to changes in the radiative budget by 2100 (e.g. see figure 12.3 of Collins et al., 2013), we chose to set the remaining (i.e. other than $CO_2$, SST, sea ice, and natural forcings) 2°C experiment forcings to their 1.5°C experiment values.

In addition to the three core experiments, modeling centers will also run at least five ensemble

members spanning the period 1959-2015, thereby allowing for a range of biases in the climate models to be assessed (see Section 4).

## 2.2  Tier 2 Experiments

The Tier 2 experiments will replicate the Tier 1 1.5 and 2°C experiments, but also take into account SST and sea ice uncertainty at the expense of ensemble size. Individual estimates of SST response

patterns from the 23 different CMIP5 models will be used, the annual means of which are presented in Appendix 1 for both scenarios. Each individual model pattern will be scaled to have the same SST mean response as the multi-model mean (MMM) response (1.02°C for the 1.5°C experiment), this would give a measure of the impact of uncertainty in the pattern of large-scale warming, conditioned on a specific global temperature change, consistent with research demanded by the UNFCCC call.

Additional Tier 2 experiments will determine the sensitivity of the response to 1.5° and 2.0°C of warming to the inclusion of atmosphere-ocean interactions in models, and hence to the choice of an AMIP-type approach for the Tier 1 HAPPI experiments. This is an important question, given that air-sea feedbacks have been shown to affect the fidelity of model representations of key phenomena that control weather and climate extremes (e.g., the Madden-Julian oscillation; DeMott et al., 2015).

These Tier 2 experiments use atmospheric GCMs coupled to either one-dimensional mixed-layer

oceans (i.e., with vertical resolution) or zero-dimensional slab oceans. These models require bias corrections to either the full vertical profile of temperature and salinity (for mixed-layer oceans) or to SST (for slab oceans) to represent missing ocean dynamics and to correct for biases in atmospheric surface fluxes (e.g., Hirons et al., 2015). A key advantage of these models is that they can maintain a given global-mean temperature effectively indefinitely. They do not include modes of coupled atmosphere-ocean variability that rely on ocean dynamics (e.g., the El Niño Southern Oscillation or the Indian Ocean Dipole), which can be an advantage as it avoids issues of under-sampling natural variability. These models are also much less computationally expensive than coupled models with full ocean GCMs.

Here, we describe the experiment design for Tier 2 experiments with the MetUM-GOML2 model, which comprises the Global Atmosphere 6.0 configuration of the Met Office Unified Model (Walters et al., 2016) coupled to the Multi-Column K Profile Parameterisation mixed-layer ocean (MC-KPP), as described in Hirons et al. (2015). First, we perform a present-day ensemble using forcing for the 1976-2005 period: greenhouse gases and aerosols are set to the average values of the period 1976-2005; temperature and salinity corrections constrain MC-KPP to the ocean climatology from Smith and Murphy (2007); and climatological sea-ice extent and concentrations are prescribed. Climatological SSTs are also prescribed in regions of seasonal sea-ice cover in the high latitudes, where the model is not coupled (see Hirons et al., 2015). 1976-2005 differs from the 2006-2015 period chosen for the Tier 1 experiments, but the objective is to understand the effect of air-sea coupling on the response to warming, not to compare the MetUM-GOML2 present-day simulation to any other model.

Secondly, we adjust the $CO_2$ in MetUM-GOML2 to achieve target global-mean warming levels, relative to the present-day ensemble, consistent with 1.5°C and 2.0°C above pre-industrial, measured by near-surface air temperature. The target levels are computed by first finding the observed global-mean surface temperature difference between 1976-2005 and pre-industrial, which is 0.52°C in HadCRUT4. The target levels are set to 1.5°C and 2.0°C minus this difference, or 0.98°C and 1.48°C, respectively. This is equivalent to projecting the change between a 1.5°C or 2.0°C warmer world and the 1976-2005 period. Finding the correct $CO_2$ concentrations involves a trial-and-error approach, but the effort is mitigated by the fact that warming is a roughly linear function of $CO_2$ (for small amounts of warming) and the model reaches steady state in 5-10 years. There are no changes to the temperature or salinity corrections, which assumes that the mean ocean heat and salt transports do not change for relatively small warming. However, we impose changes to sea ice and the prescribed SSTs in uncoupled (seasonally ice-covered) regions. We compute these using a transient simulation of the fully coupled MetUM-GC2 (Williams et al., 2015) with a 1% yr-1 $CO_2$ increase, by averaging 20-year periods with global-mean warming closest to our 0.98°C and 1.48°C target levels and taking the difference between these periods and the climatology of the MetUM-GC2 present-day control simulation. We apply these differences to the 1976-2005 observed climatologies.

Thirdly, we perform initial condition perturbation ensembles of MetUM-GOML2 simulations at the target warming levels, using the $CO_2$ concentrations and sea-ice and high-latitude SST boundary conditions determined above. Finally, we perform AMIP-type experiments with the same atmospheric model, in which we prescribe the daily SSTs and sea-ice from the MetUM-GOML2 ensembles. MetUM-GOML2 uses a 3-hr coupling frequency; converting to daily SSTs introduces sufficient noise to cause the coupled and atmosphere-only experiments to diverge.

Comparing the coupled and AMIP-type experiments at the same level of warming allows one to determine the sensitivity of the response to the presence of atmosphere-ocean interactions, in a framework in which the mean and inter-annual variability of SST and sea ice are consistent between the simulations. Similarly, comparing the relative difference between the 1.5°C and 2.0°C simulations in the coupled and AMIP-type experiments allows one to determine whether the response to an additional half-degree of warming is sensitive to inclusion of air-sea coupled feedbacks. We expect that analysis of these experiments will focus mainly on sub-seasonal variability and extremes (e.g., heatwaves, intense precipitation events), but it is possible that air-sea coupling will also affect the mean response.

## 3 Toward understanding impacts

Assessing potential impacts of 1.5 and 2°C of warming goes beyond climate scenarios and requires integrated impact model projections. HAPPI therefore cooperates with the Inter-Sectoral Impact Model Intercomparison Project (ISIMIP, Warszawski et al., 2013a) range of sectors including agriculture and agro-economic modeling (Rosenzweig et al., 2013; Elliott et al., 2013), water (Schewe et al., 2014), biomes and forestry (Warszawski et al., 2013b), permafrost and human health (Mitchell et al., 2016a). To allow for the HAPPI modeling effort to be most useful for the impact community, the HAPPI diagnostics provided resemble the climate model input required for the ISIMIP modeling protocol.

Specifically, a priority subset of HAPPI AGCM output will be provided in bias-corrected format following the ISIMIP2b bias correction approach (Frieler et al., 2016). A sector specific modeling protocol will be available following the ISIMIP2b simulation protocol including socio-economic and management options.

## 4 Data usage and availability

Data published on the portal will be compliant with a modified version of the C20C+ D&A conventions. All raw data will be available, as well as a bias corrected ISIMIP subset using the Frieler et al. (2016) methodology.

Output from all HAPPI and associated experiments are to be published through the joint C20C+ D&A project-HAPPI portal, hosted by the National Energy Research Scientific Computing center

(NERSC) at $http://portal.nersc.gov/c20c/data.html$. The HAPPI data policy uses the same principles as the Coupled Chemistry Model Validation (CCMVal) policy. The HAPPI data are therefore made available to all researchers outside the HAPPI community, provided that they become official HAPPI collaborators. All collaborators are asked to respect the interests of the HAPPI community, and therefore encouraged to keep lines of communication throughout any analysis. Publications of HAPPI data and corresponding scientific analysis are encouraged, and the data policy involves two phases in line with CCMVal. Phase 1 runs up to the cut-off date for publications to be included in the IPCC Special Report (in April 2018). During this phase users are obligated to offer co-authorship to the HAPPI core-team, and to acknowledge NERSC for data storage. Phase 2 follows publication of the IPCC Special Report, and requires acknowledgment of the HAPPI core-team and NERSC. During the latter phase it is intended that HAPPI data will be used to inform AR6 among other initiatives, and may well include high temperature scenarios, such as $3°$C.

## 5 Summary

HAPPI has been developed to explicitly inform one of the primary aims in the Paris Agreement, which seeks to understand impacts of a world limiting global averaged warming to $1.5°$C. It provides climate data for analysis of a range of impacts under current, 1.5 and $2°$C climate scenarios. The high number of ensemble members (>50) allow for information on policy-relevant timescales to be assessed, while the 10-year length of the simulations also allows for long-lived extremes, such as droughts, to be characterized. The two tiers of experiments provide an assessment of not only the desired climate change scenario, but also the uncertainties in how we developed the scenario, most notably through sensitivity tests in the SSTs and sea ice conditions. The data are available in bias corrected or raw formats, and ready for direct input to a range of common climate impact models.

**HAPPI website:** The project is kept up-to-date with news, collaborations, publications and experiments at www.happimip.org.

*Acknowledgements.* We would like to thank Ben Sanderson, Reto Knutti and Annette Hirsch for their in depth reviews of our paper. DMM received support from the NERC ACE-Africa and a NERC Independent Research Fellowship (NE/N014057/1). JSF, IB, TI and OS received support from the Norwegian Research Council, project no. 261821. This material involved work supported by the U.S. Department of Energy, Office of Science, Office of Biological and Environmental Research, under contract number DE-AC02-05CH11231. HS was supported by the Program for Risk Information on Climate Change from the Ministry of Education, Culture, Sports, Science and Technology of Japan, and by the Environment Research and Technology Development Fund (S-10) of the Ministry of the Environment of Japan. CFS was supported by the German Federal Ministry for the Environment, Nature Conservation and Nuclear Safety (11_II_093_Global_A_SIDS and LDCs).

NPK was funded by an Independent Research Fellowship from the UK Natural Environment Research Council (NE/L010976/1).

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

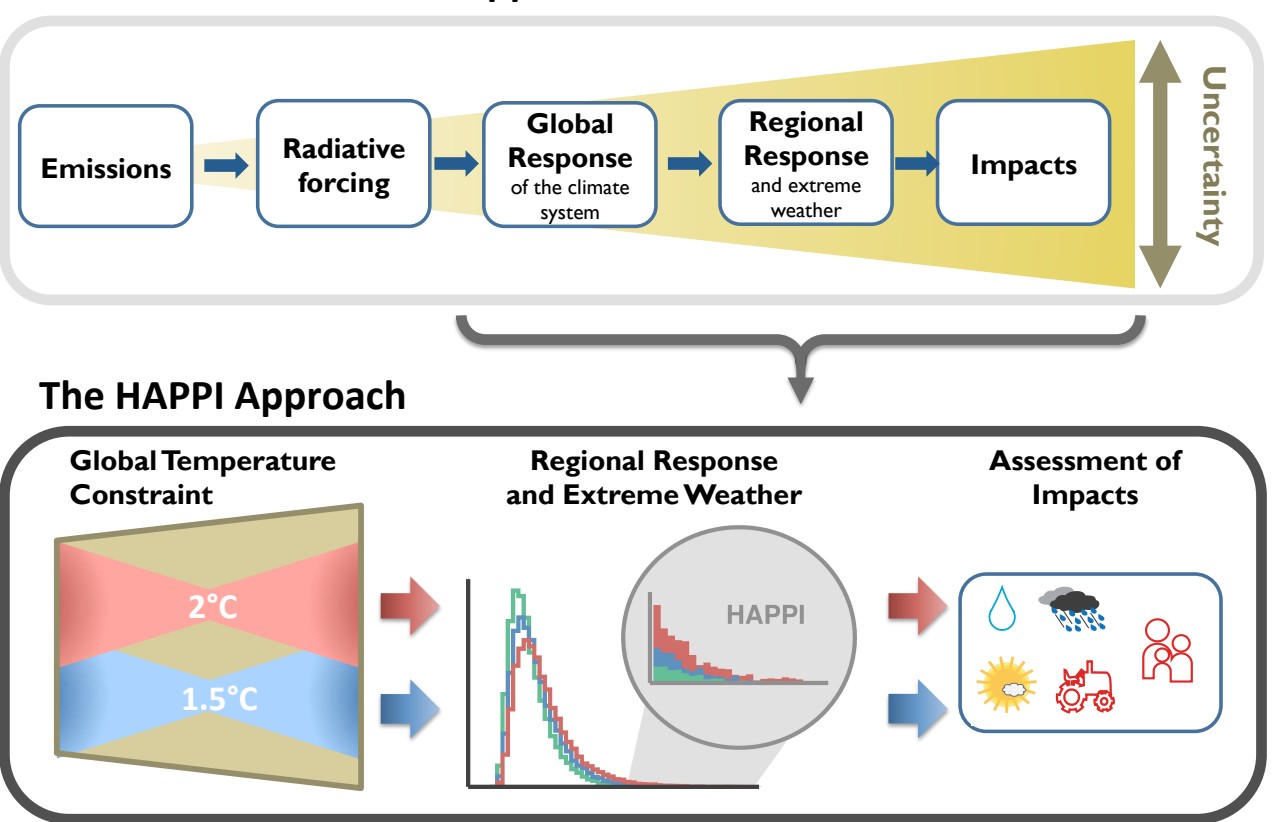

Figure 1: A schematic comparing the emissions scenario based approaches (top), such as CMIP, with the HAPPI approach (bottom). The HAPPI approach flows from the constraint on global temperatures to the comparison of extremes using the large ensemble approach to impact models. The histogram depicts an illustrative example of distributions for extreme event indicators (such as e.g. maximum daily temperature) for the present day (green), 1.5°C (blue) and 2°C (red) above pre-industrial levels

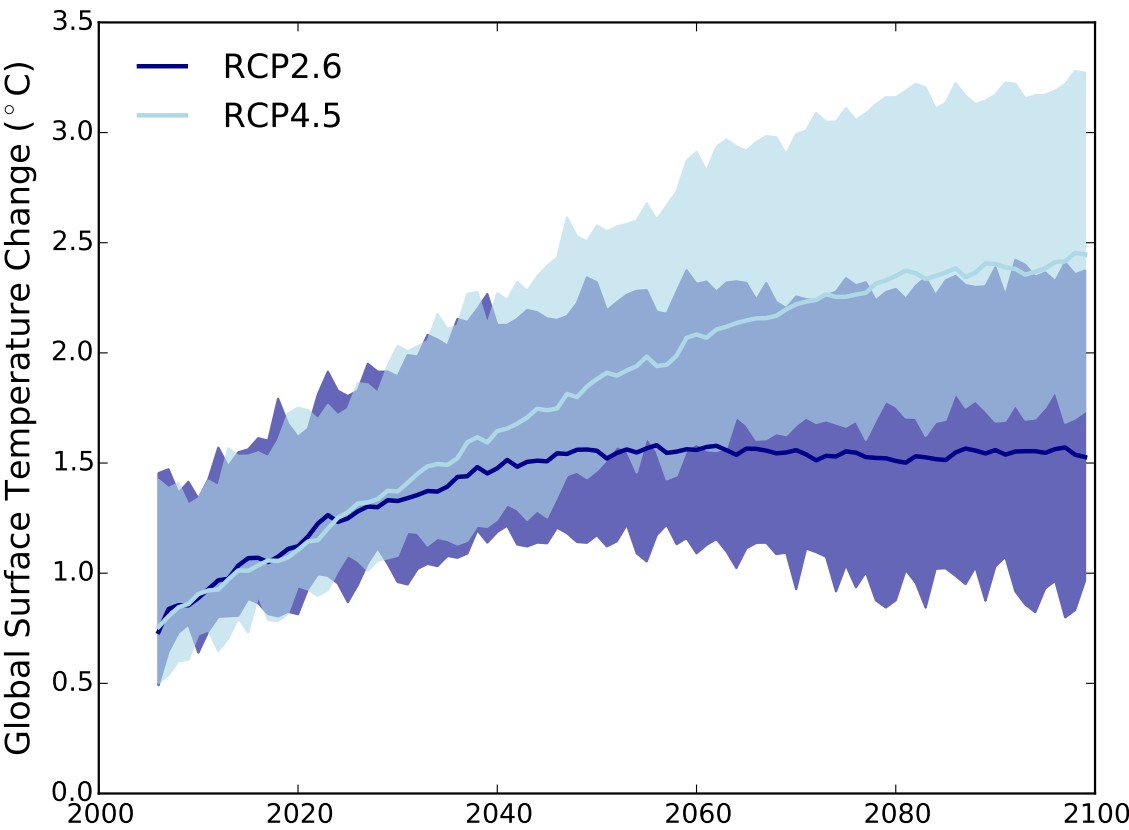

Figure 2: Time series of global annual mean surface air temperature anomalies (relative to 1861-1880) from CMIP5 RCP2.6 and RCP4.5 experiments. Solid lines show the multi-model mean and shaded regions show the 5-95% range across all 26 models. Only one simulation is used for each model. All models where the data were available for both scenarios were used, leading to 26 models in total.

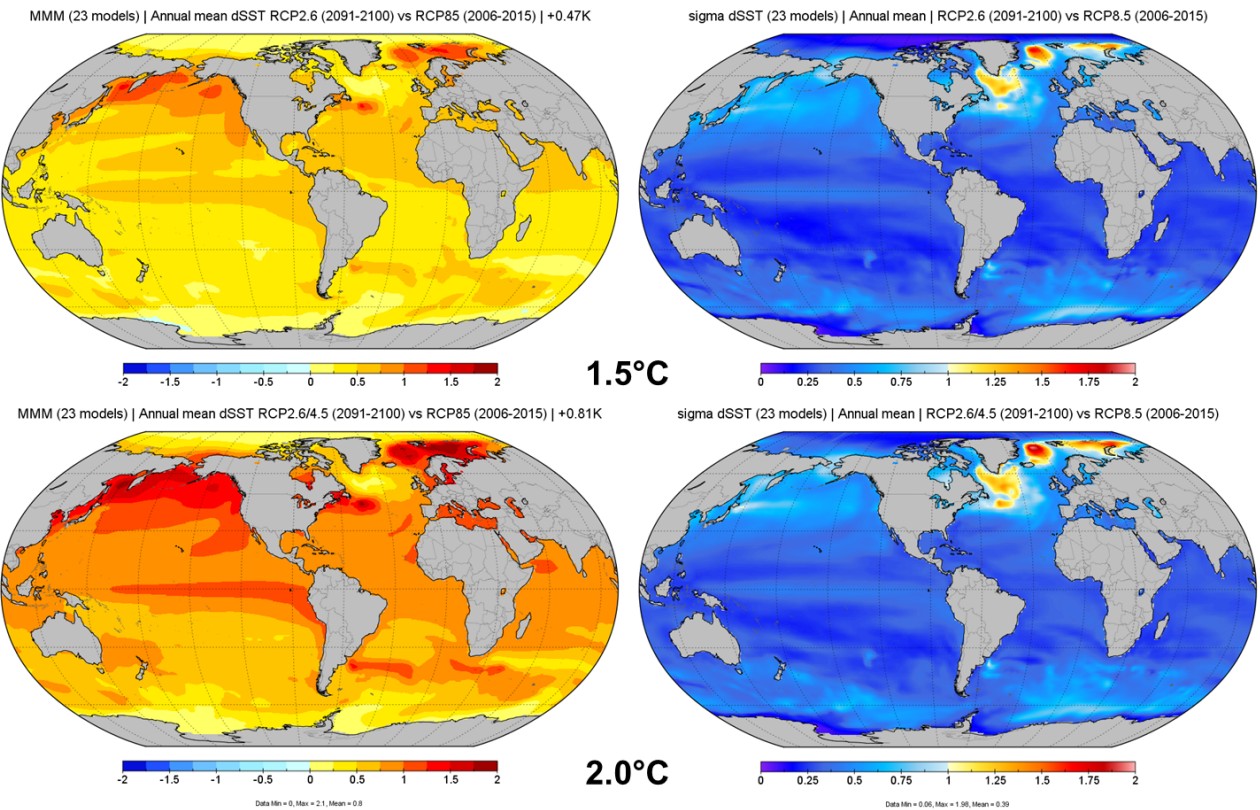

Figure 3: (left) SST warming pattern added to the current decade to produce the (top) 1.5 and (bottom) 2 degree scenarios. (right) The standard deviation of annual mean delta SSTs across the 23 models. Units are in °C.

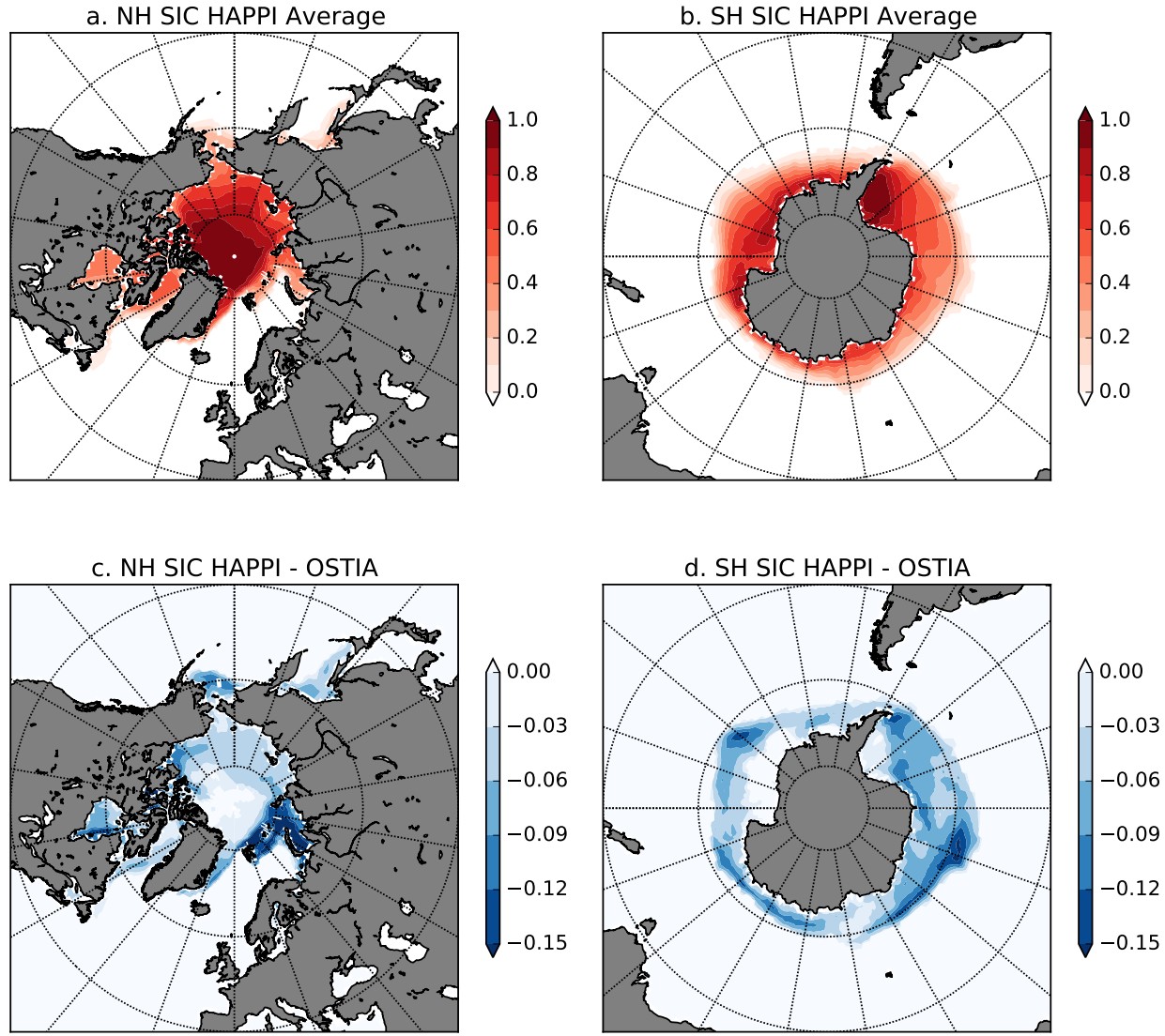

Figure 4: Polar stereographic projections of decadal-mean (top) sea ice concentration from the 1.5 degree experiment and (bottom) the difference in sea ice concentration between the 1.5 degree experiment and OSTIA. The OSTIA data cover the decade 2006-2015. Left panels show the NH, right panels show the SH.

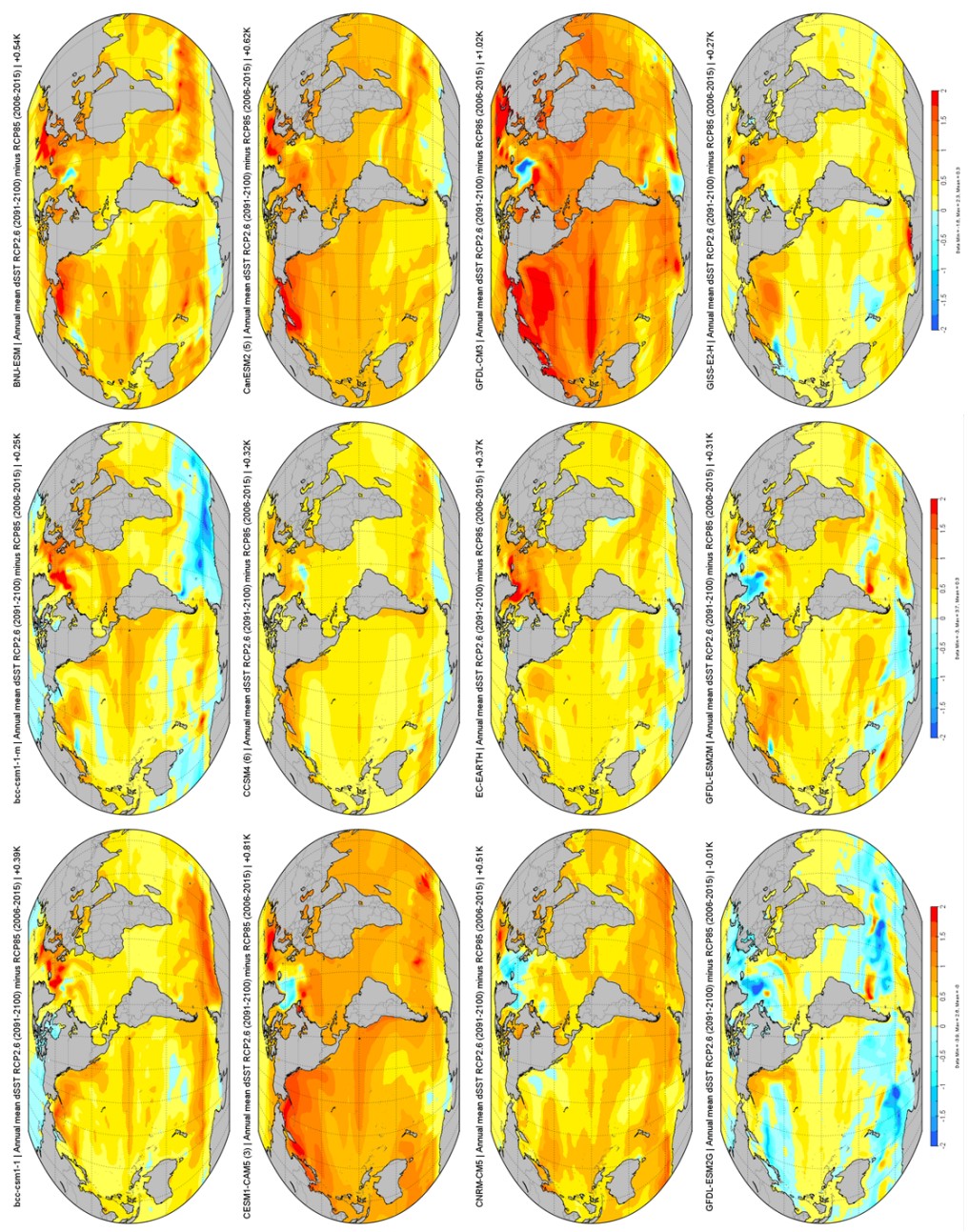

Figure A1: As in the 1.5 degree experiment delta SST pattern in Fig. 3 but for the first set of 12 individual models.

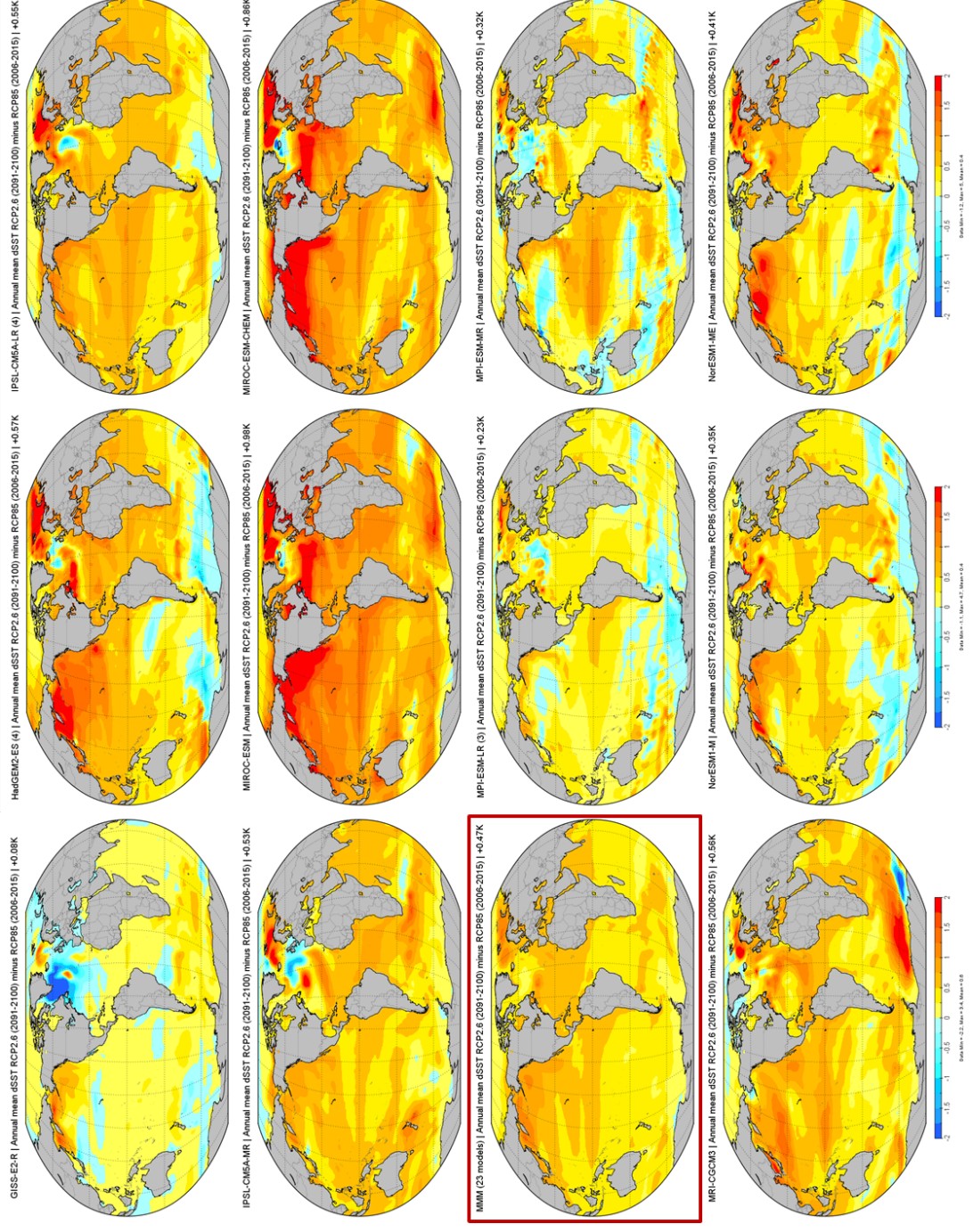

Figure A2: As previous but for the second set of 12 individual models.