# Peer review of "Half a degree Additional warming, Prognosis and Projected Impacts (HAPPI): Background and Experimental Design"

_Geoscientific Model Development, 2016_

## Referee Comment (RC1) · B. Sanderson (Referee) · 14 Sep 2016

The submitted paper details a proposal for HAPPI - a model inter-comparison project designed to respond to the request in the 2015 Paris Agreement for the IPCC to assess climatological differences between futures under which the world underwent a 1.5 or 2 degree warming above pre-industrial values. The paper outlines an experimental protocol in which AMIP experiments are conducted to represent the recent past, and futures corresponding to the two levels of warming. The AMIP experiment for the recent past is a 10 year experiment under a standard AMIP protocol, whereas the future experiments use anomaly fields derived from multi-model averages from the CMIP5 RCP experiments. The 1.5 degree future is achieved using the CMIP5 multi-model

average anomaly sea surface temperature fields from the RCP2.6 experiment (2091-2000 vs 2006-2015), which is added to the historical 2006-2015 SSTs. The 2 degree experiment uses a linear combination of RCP2.6 and RCP4.5 SSTs, which is predicted to produce a warming of .5K greater than the 1.5 degree experiment. Non-CO2 forcings and land use are kept constant in the two experiments using the RCP2.6 values for both. Sea ice concentrations will be predicted through a linear regression approach, relating historical SSTs to Sea Ice concentration.

The paper makes a strong case for why this experimental design will provide useful data for the IPCC Special Report on 1.5 degrees. The existing CMIP5 ensemble experiments do not provide a clean answer to the question posed for the report (that of how a 1.5 and 2 degree climate are likely to differ), and the experimental design proposed for HAPPI will certainly provide a well sampled estimate of the climatology associated with these two warming levels. My conclusion is that the paper, and concept are broadly sound - and that the paper and subsequent ensemble will be an asset for the IPCC in their preparation of the special report, in conjunction with other data. I have some minor comments on the technical details of the implementation but see no major flaws in the article.

Minor Comments:

1 - How the SSTs anomalies are created is currently ambiguous. The text implies the anomalies are the difference between 2006-2015 and 2091-2100 in RCP2.6 (or 4.5). However, Figure 3 would suggest that RCP8.5 is used as the recent period anomaly baseline. The authors should make this clear.

2 - The selection of 2006-2015 as the base period is convenient, as the authors note because it represents a 'stable' recent climate. However, it might also prove troublesome in exactly recreating a 1.5 degree and 2 degree warming because it's likely that the hiatus years bracketed by this period are likely cooler than the climatological attractor (Meehl et al 2016, Trenberth Fasullo, 2013). However, the RCP average for

2006-2015 will represent the climatological attractor. As such, one would expect that the 1.5 and 2 degree SST reconstructions will be biased cold when compared to the CMIP5 multi-model average from which they were derived. The fact that the CMIP5 average temperature for RCP2.6 for 2091-2100 is actually 1.55K above pre-industrial might actually compensate for this bias, but the authors should probably at least discuss the implications of using the recent hiatus period as the baseline.

3 - The decision to use a linear regression to produce sea ice distributions is well defended (or at least a case is made for why it would be inappropriate to use the RCP model sea ice distributions directly), but whether the regression will actually work or not is conditional on exactly why the current generation of models is apparently biased in its Antarctic sea ice distribution. If the discrepancy between recent historical Antarctic sea ice behavior is due to non-temperature mitigated variability (as suggested by Turner et al, 2015), then such a regression-based approach might produce an under-estimate of the likely sea reduction in the antarctic at the end of the century. Perhaps the protocol could include a second tier sensitivity study to assess how the conclusions might be impacted if the authors instead took a more model-centric view of future sea-ice change.

4 - Framing the experiment as a single 10 year period with anomalies added carries a risk that the significance of the difference in the two climate states is going to be overestimated. Because every simulation in the ensemble will undergo the same SST variability, the difference in the resulting climate arising from the anomaly pattern differences will be put in the context of atmospheric noise only. This is, of course, an underestimate of the natural variability which would be present if ocean states were also sampled - and as such, there is a concern that studies based on this ensemble might over-state the differences in climate risks between the two temperature levels. It is clear enough why the AMIP-based approach conveys some advantages in its ability to address the Paris Agreement's request, but unless the space of SST variability is sampled - there is a danger that studies based on these ensembles will tend to con-

clude that differences between the 2 degree and 1.5 degree climate states are more significant than they would actually be in reality.

This concern could be addressed using another second tier experiment. Whereas the tier-2 experiments in the current version focus on uncertainty in the anomaly pattern - I would see some stronger logic in using that computing time to sample the impact of SST variability. A simple experiment could be to replace the 2006-2015 baseline with RCP SSTs from individual models, then adding the same anomaly as is used in the original experiments. As such, one could create an ensemble of constructed 2091-2100 periods with different SST variability - sampling the range of possible modes of varaibility which could potentially define that decade.

5 - the choice to only change the CO2 forcing between the 1.5 degree and 2 degree simulations may cause the difference between the two simulations to be less than 0.5K. The decision to not change land use or aerosols between the two simulations is understandable - and the net radiative forcing difference between RCP2.6 and RCP4.5 for these agents is unlikely to be significant. However, this is not true of non-CO2 greenhouse gases (CH4, N2O  CFCs) which do differ between RCP4.5 and RCP2.6 and will likely have a significant impact on the net radiative forcing. Furthermore, because these are well mixed gases - it would be easy enough to produce consistent concentrations in a similar manner to the CO2 calculation.

I do not consider any of these issues to be fatal flaws, but look forward to the authors' thoughts. Furthermore, I look forward to seeing the data and results arising the experiment - which should be an asset to the international climate community,

Ben Sanderson

References

Meehl, Gerald A., Aixue Hu, Benjamin D. Santer, and Shang-Ping Xie. "Contribution of the Interdecadal Pacific Oscillation to twentieth-century global surface temperature

trends." Nature Climate Change (2016).

Trenberth, Kevin E., and John T. Fasullo. "An apparent hiatus in global warming?." Earth's Future 1, no. 1 (2013): 19-32.

Turner, John, J. Scott Hosking, Thomas J. Bracegirdle, Gareth J. Marshall, and Tony Phillips. "Recent changes in Antarctic Sea ice." Philosophical Transactions of the Royal Society of London A: Mathematical, Physical and Engineering Sciences 373, no. 2045 (2015): 20140163.
* * *

---

## Referee Comment (RC2) · R. Knutti (Referee) · 28 Sep 2016

Summary:

This paper describes the setup of a series of AMIP style fixed SST large ensemble simulations with different climate models, in order to inform in particular the changes in extreme weather events for today, for 1.5 and 2°C. The results are targeting the impacts community and should feed into the IPCC special report on the 1.5 and 2°C climate target.

Review:

I welcome the proposed effort to inform the upcoming IPCC special report with this

targeted experiment. This is both timely and relevant and will complement other efforts like ScenarioMIP within CMIP6 that will not happen before the deadlines imposed by the special report. The public availability of the data will ensure that other communities can benefit from that.

I have no major comments on this paper, since it's just a description of an experimental setup that is, in its core, not much different from earlier setups using fixed SSTs. The setup is likely to work fine, whether the results will provide a major step forward remains to be seen. The results might be unsurprising from a climate point of view in that most things appear to scale rather well with global temperature, but that may not apply to impacts or to specific regional questions. In any case it is worth doing this experiment, as it is straightforward yet interesting and valuable for both science any policy. The first reviewer has already made many important points which I support. I only have a few more comments below that the authors should discuss in a revised version.

The statement that the classic emission scenario approach is problematic to infer impacts for certain warming levels (line 51) is assuming that we look at the projections for a specific time period. But one can simply pick the 20yr period in which a particular model reaches 1.5°C or 2°C and aggregate that (as done for example in Fischer and Knutti 2015). That is assuming that the patterns in a transient 1.5°C world are similar to those in a near equilibrium 1.5°C world, but making that assumption is unlikely to introduce large biases, at least compared to the uncertainty of the warming pattern itself (e.g. Herger et al. 2014). This would on the other hand have the advantage of sampling results from the fully coupled model, including different patterns of SST and representations of coupled ocean atmosphere variability, which HAPPI cannot do. Of course it is more expensive in terms of computing and provides less model years, but given over a hundred CMIP5 ensembles it is still informative, and fundamentally I don't see why such an approach would be "very difficult" (line 51). In my view the two are complementary, and I think the current wording could be improved to be more balanced.

I'd like the authors to comment on using a decadal mean SST as a boundary condition vs. time varying fields. Is there a problem of say suppressing El Nino events by fixing the SST at a long term average, and could that have an effect on the frequency of extremes? What if the magnitude or timescale of ENSO changes as a result of warming, how would that affect changes in extremes in the regions that are affected by ENSO teleconnections, and would the proposed setup account for that? It seems like testing different SST patterns will sample some uncertainty but coupled variability would not be addressed by that.

Reto Knutti

–

References:

Fischer, E. M., and R. Knutti. 2015. "Anthropogenic Contribution to Global Occurrence of Heavy-Precipitation and High-Temperature Extremes." Nature Climate Change 5 (April): 1–6. doi:10.1038/nclimate2617.

Herger, Nadja, Benjamin M. Sanderson, and Reto Knutti. 2015. "Improved Pattern Scaling Approaches for the Use in Climate Impact Studies." Geophysical Research Letters 42 (9): 3486–94. doi:10.1002/2015GL063569.

---

## Short Comment (SC1) · 19 Oct 2016

This paper provides a description of the HAPPI MIP simulation protocol. While the simulation design is ambitious, this is a very timely experiment as results from HAPPI MIP will inform the IPCC special report on 1.5°C and provide the community with a broader sense of the possible difference in impacts between 1.5°C and 2.0°C.

I have one comment about the protocol design concerning the land-use description. In particular, it would be useful to have an explicit statement on the treatment of land cover for both 1.5°C and 2.0°C experiments. Land use and land cover change has been shown to have a substantial effect on regional climate as demonstrated in LUCID (e.g. Pitman et al., 2009) and will be examined further in LUMIP (Lawrence et al., 2016).

[Figure]

Currently it is not sufficiently clear whether participants should use fixed land cover from RCP2.6 for the year 2100 or the decadal average (over 2106-2115) or whether the is land cover the same in both 1.5°C and 2.0°C experiments? This is implicitly implied towards the end of Section 2.1 (line 158) but perhaps having a clear statement would avoid any ambiguity.

References

Pitman, A. J., and Coauthors, 2009: Uncertainties in climate responses to past land cover change: First results from the LUCID intercomparison study. Geophysical Research Letters, 36, L14814, doi:10.1029/2009GL039076.

Lawrence, D. M., and Coauthors, 2016: The Land Use Model Intercomparison Project (LUMIP) contribution to CMIP6: rationale and experimental design. Geosci. Model Dev, 9, 2973–2998, doi:10.5194/gmd-9-2973-2016.

---

## Author Comment (AC1) · 16 Dec 2016

Summary

We thank the reviewers for their nice assessment of our paper. All reviewers suggested minor corrections, and we have responded to all of them, making most suggested changes. A running theme was in expanding the discussion on various choices of model forcing that we have chosen. This has been done, with a specific focus on how fixed SST experiments may alter extreme events compared with coupled-ocean experiments.

Reviewer 1

1 - How the SSTs anomalies are created is currently ambiguous. The text implies the anomalies are the difference between 2006-2015 and 2091-2100 in RCP2.6 (or 4.5). However, Figure 3 would suggest that RCP8.5 is used as the recent period anomaly baseline. The authors should make this clear. We use the RCP8.5 scenario for the 2006-2015 baseline, as this is the closest to observations. This has now been made clearer in the paper. L128-130. 2 - The selection of 2006-2015 as the base period is convenient, as the authors note because it represents a 'stable' recent climate. However, it might also prove trouble- some in exactly recreating a 1.5 degree and 2 degree warming because it's likely that the hiatus years bracketed by this period are likely cooler than the climatological at- tractor (Meehl et al 2016, Trenberth Fasullo, 2013). However, the RCP average for 2006-2015 will represent the climatological attractor. As such, one would expect that the 1.5 and 2 degree SST reconstructions will be biased cold when compared to the CMIP5 multi-model average from which they were derived. The fact that the CMIP5 average temperature for RCP2.6 for 2091-2100 is actually 1.55K above pre-industrial might actually compensate for this bias, but the authors should probably at least dis- cuss the implications of using the recent hiatus period as the baseline. We have now added in this short discussion, but we note that it is the increase of a 0.5K warming in low emissions scenarios that is important, rather than the absolute numbers. L134-139. 3 - The decision to use a linear regression to pro- duce sea ice distributions is well de- fended (or at least a case is made for why it would be inappropriate to use the RCP model sea ice distributions directly), but whether the regression will actually work or not is conditional on exactly why the current generation of models is apparently biased in its Antarctic sea ice distribution. If the discrepancy between recent historical Antarctic sea ice behavior is due to non-temperature miti- gated variability (as suggested by Turner et al, 2015), then such a regression-based approach might produce an under-estimate of the likely sea reduction in the antarctic at the end of the century. Perhaps the proto- col could include a second tier sensitivity study to assess how the conclusions might be impacted if the authors instead took a more model-centric view of future sea-ice change. We feel that every choice of forcing

in our setup could be tried and tested through sensitivity analyses, and some of this will be performed by individual modelling centres. However, for this particular case we feel that it is not worth dedicating space in the experimental design, as the same could be said for many other equally important components of our experimental design. Instead we concentrate on the fixed SSTs as the reviewer suggests below. 4 - Framing the experiment as a single 10 year period with anomalies added carries a risk that the significance of the difference in the two climate states is going to be overestimated. Because every simulation in the ensemble will undergo the same SST variability, the difference in the resulting climate arising from the anomaly pattern dif- ferences will be put in the context of atmospheric noise only. This is, of course, an underestimate of the natural variability which would be present if ocean states were also sampled - and as such, there is a concern that studies based on this ensemble might over-state the differences in climate risks between the two temperature levels. It is clear enough why the AMIP-based approach conveys some advantages in its ability to address the Paris Agreement's request, but unless the space of SST variability is sampled - there is a danger that studies based on these ensembles will tend to conclude that differences between the 2 degree and 1.5 degree climate states are more significant than they would actually be in reality. This concern could be addressed using another second tier experiment. Whereas the tier-2 experiments in the current version focus on uncertainty in the anomaly pattern - I would see some stronger logic in using that computing time to sample the impact of SST variability. A simple experiment could be to replace the 2006-2015 baseline with RCP SSTs from individual models, then adding the same anomaly as is used in the original experiments. As such, one could create an ensemble of constructed 2091- 2100 periods with different SST variability - sampling the range of possible modes of varaibility which could potentially define that decade. We agree with the reviewer on this point, and have added an additional experiment set under Tier 2 to reflect this. This experiment uses another model (MetUM-GOML2) to run coupled ocean experiments and fixed SST experiments in a fashion that allows for a direct comparison. As this is a new experimental design, we have dedicated a large section to

explaining it. L194-250. 5 - the choice to only change the CO2 forcing between the 1.5 degree and 2 degree simulations may cause the difference between the two simulations to be less than 0.5K. The decision to not change land use or aerosols between the two simulations is un- derstandable - and the net radiative forcing difference between RCP2.6 and RCP4.5 for these agents is unlikely to be significant. However, this is not true of non-CO2 greenhouse gases (CH4, N2O CFCs) which do differ between RCP4.5 and RCP2.6 and will likely have a significant impact on the net radiative forcing. Furthermore, be- cause these are well mixed gases - it would be easy enough to produce consistent concentrations in a similar manner to the CO2 calculation. Yes, this was originally missed from the paper. All well-mixed GHGs are scaled as the reviewers suggests. L169-173.

---

## Author Comment (AC2) · 16 Dec 2016

The statement that the classic emission scenario approach is problematic to infer impacts for certain warming levels (line 51) is assuming that we look at the projections for a specific time period. But one can simply pick the 20yr.

"We have expanded on our discussion here to reflect the reviewers comments. L51-62."

I'd like the authors to comment on using a decadal mean SST as a boundary condition vs. time varying fields. Is there a problem of say suppressing El Nino events by fixing the SST at a long term average, and could that have an effect on the frequency of extremes?

"We do not use a decadal mean SST, just a delta SST decadal mean. This is added to observed SSTs which still contain the time varying component. This was perhaps not clear in the initial text, so has been reworked. L126-127."

What if the magnitude or timescale of ENSO changes as a result of warming, how would that affect changes in extremes in the regions that are affected by ENSO teleconnections, and would the proposed setup account for that? It seems like testing different SST patterns will sample some uncertainty but coupled variability would not be addressed by that.

"We agree with the reviewer on this point, and have added an additional experiment set under Tier 2 to reflect this. This experiment uses another model (MetUM-GOML2) to run coupled ocean experiments and fixed SST experiments in a fashion that allows for a direct comparison. As this is a new experimental design, we have dedicated a large section of the paper explaining it. L194-250."

―――――――――――――――

---

## Author Comment (AC3) · 16 Dec 2016

We appreciated the importance of land use and land cover, and information on the specifics has been updated to reflect this. L121-122.
* * *